# Application of 3-D Drucker–Prager Material Model to Determine Optimal Operating Parameters of Centrifugal Regeneration Device

**DOI:** 10.3390/ma13092134

**Published:** 2020-05-05

**Authors:** Waldemar Łatas, Rafał Dańko, Przemysław Czapla

**Affiliations:** 1Institute of Applied Mechanics, Faculty of Mechanical Engineering, Cracow University of Technology, Al. Jana Pawła II 37, 31-864 Kraków, Poland; waldemar.latas@pk.edu.pl; 2Faculty of Foundry Engineering, AGH University of Science and Technology, Al. A. Mickiewicza 30, 30-059 Kraków, Poland; 3Eurocast Industries, ul. Grabowa 19, 30-227 Kraków, Poland; pczapla@eurocast-ind.com

**Keywords:** molding sand mechanical reclamation, centrifugal regeneration unit, circular economy, Drucker-Prager yield criterion, Material Point Method

## Abstract

The process of metal casting indisposable sand molds is associated with the generation of large amounts of waste, mainly used molding and core sands, from which the molds and cores reproducing the external and internal shapes of the castings were made. It is estimated that about 600 kg of waste can come from the production of 1 ton of casting. The main component of the waste is quartz matrix, which after undergoing appropriate reclamation treatments can be recovered and reused in the production process. This article presents the theoretical foundations regarding the existing methods of quartz matrix recovery and an experimentally justified model of the regeneration process occurring in one of the varieties used in the practice of mechanical regenerators. The goal is to improve the quality of regenerated molding sand by means of liberating the sand grain’s surface from the layer of the used binding component. The elastic-plastic material model characterized by the Drucker–Prager yield criterion was used to describe the deformation of the sand layer during treatment performed in a centrifugal regenerator. Conclusions based on the results of numerical calculations, obtained with the use of the software adopting the material point method, enable us to find out how to control the device in a way that ensures a permanent reclamation effect which is independent of the working components that wear out over time.

## 1. Introduction

In the process of manufacturing cast metals (mainly of iron and non-ferrous alloys) in single-use sand molds made of chemically bonded molding sands, the external and internal shapes of the casting are reproduced by the mold and core. The material that creates these elements is quartz sand with appropriate properties, the individual grains of which are joined together with a binder to form a mold or casting core [1,2]. The quartz sand used in the mold and core production process must have adequate grain size, main fraction content, homogeneity, proper shape and heat resistance [3]. Molds and casting cores are made in such a way that their shapes ensure the correct filling of the interior of the mold with liquid metal and the creation of a casting free of internal and external defects with an appropriate, previously established microstructure. It is assumed that in the case of traditional technologies, the amount of material used to make the mold and cores (quartz sand + binder) is 3.5–5 times greater than the amount of metal poured into the mold for casting [2,4,5]. After the casting is made, it is separated from the casting mold in the stamping out or knocking out processes and subjected to further processing. The molding and core sands formed after separation constitute waste that can undergo the regeneration process [2,6].

The most important issues for the considered topic of regeneration of used molding and core sands were analyzed in [7,8,9,10,11,12,13,14,15,16,17,18,19]. To date, these items bring the most complete overview on many aspects of this process, its variations and applications, which have been ordered and classified into specific topics and specific issues. The general issues related to the regeneration of used sands in the context of the process variants applied are highlighted below.

Publications [7,8,9,10,11,12,13,14,15] treated the mechanical regeneration processes as specific variations of the method of releasing sand matrix from the coating of used binding material that occur regardless of the type of regenerative treatment. Practically to date, no model has been developed that allows a quantitative description of the phenomena in the process of releasing matrix grains from the coating of binding material. In publications [16,17], attempts were made to determine the mechanisms of destruction of binder bonds during the thermal impact on the regenerated matrix. However, their application in the regeneration process does not allow a quantitative description. The analytical model of this process [18] based on the crushing theory is also known, but it was not a numerical solution to the problem, which is demonstrated in the presented work. The distinction in the process of regenerative treatment of two of its ranges (pre-regeneration and specific regeneration) emphasized in the literature [19] has a conventional nature useful for the certain systematics of the analyzed phenomena, both at the boundary between the binding material and the matrix grain, as well as in the contact area of individual grains in spatial terms.

The analysis of the available literature indicates that no attempt has been made so far to develop the numerical model of the mechanical regeneration process, which is presented in this publication.

The model calculations and verification tests presented in the publication concern one of the varieties of dry mechanical (proper) regeneration device, namely the centrifugal regenerator. The goal was to determine a control method of the device’s work which, by proper selection of the roller pressure on the layer of sand located on the internal side surface of the regenerator batch, will ensure a constant regenerative effect of used mass grains, independent of its working elements wearing out in time.

The elastic-plastic material model characterized by the Drucker–Prager yield criterion [20] was used to describe the deformation of the sand layer. The regeneration quality indicator related to shear strain was defined, and, on its basis, the value of the optimal regenerator supply pressure was obtained depending on the depth of the groove of the device batch’s side surface. In the numerical calculations, a program based on the material point method [21] was used.

## 2. Regeneration of Used Molding and Core Sands

Regeneration of used molding and core sands is defined as a treatment that allows the recovery of at least one of the components of these sands with properties similar to the fresh state of this component and its re-use for the production of molds and cores. Currently, the main component recovered from used molding and core sands is quartz sand, constituting the matrix and the main component of these sands.

Used molding and core sands can be used alone or in conjunction with other components in the construction industry, as a sealing material in landfills, as a material for the production of cement aggregate, or asphalt filler [3,4,5,22]. In the foundry, the sands can be pre-prepared for non-foundry applications by subjecting them to a pre-regeneration process. However, the optimal way to use the sands is to treat them so that they can be reused for molds and cores. Possible ways of dealing with used sand are shown schematically in Figure A1 (Appendix A).

Used sand, regardless of the adopted method of matrix recovery from it, is subjected to treatment carried out in appropriate systems, most often with the following sequence of actions:preliminary separation of mechanical impurities from the used sand, mainly metal onesgrinding the compacted sand after knocking it out of the moldsieving the sand and isolating the grain size range of the material for regenerationre-separation of metal impurities from the sandproper (secondary) regeneration, releasing the matrix from the remains of the used binding material by using methods that allow removal of a coating of binding material from the surface of the grainsremoving unwanted regeneration products from the matrix by dedusting itisolation of a matrix with a specific grain size and homogeneity (classification by grain size)

The treatment of the used sand consisting of the combination of the first four mentioned activities is referred to as the primary regeneration process [2,3]. Its full implementation is a prerequisite for subjecting the obtained matrix to subsequent, final treatment of the matrix grain purification from the coatings of binding material (proper regeneration—named secondary regeneration). The process of secondary regeneration should lead to obtaining a product (quartz sand, matrix) which is characterized by properties similar to the fresh state of this component and can be its substitute in the process of manufacturing molds and cores.

Figure A2 (Appendix A) presents a schematic view of the proper (secondary) regeneration methods used, with the basic classification criterion being the environment in which the regeneration process is carried out. In this aspect, two basic wet and dry methods can be distinguished.

The dry regeneration process by mechanical means has today found the widest application in foundries, which results from the following possibilities:matrix recovery from virtually any used sand, assuming a limited degree of matrix cleaning from residual binding material (regeneration rate)applications for the regeneration of relatively simple devices, often for other original purposes (for example mixers)lower, compared to other methods, costs of the regeneration process

It seems that the variety of regenerative interactions used in mechanical regeneration devices should be considered through the prism of the so-called elementary operations that share common characteristics regardless of the treatment method. This approach was presented in [2,3].

The purpose of the mechanical regeneration process is to affect the grain of used sand (sand matrix surrounded by a layer of used, inactive binder) in such a way as to separate the used binder from the matrix (grain) and thus clean the matrix grain to a degree similar to that of the matrix when fresh. This process is implemented through the so-called elementary operations that are carried out in devices called regenerators [11,23]. The elementary operations occurring in mechanical regeneration—rubbing, abrasion and crushing—are presented in Figure A3, Figure A4 and Figure A5 and described in more detail in Appendix A.

Model calculations and verification tests presented in the publication concern one of the varieties of devices for mechanical dry regeneration.

## 3. Reclamation Device Preliminary Tests Results

The USR-I reclamation device was used in model, preliminary and verification tests (Figure A6, Appendix B). The device is a centrifugal regenerator with a periodic work cycle and of a relatively universal purpose, equipped with additional grinding and compressing components placed in the batch. The device batch and other internal structural components exposed to intense abrasion are made of abrasion-resistant ceramic materials.

The portion of used sand that can be at one time subjected to reclamation treatment in the device is 25–30 kg. The regenerator is equipped with a timer which makes it possible to adjust the duration of reclamation treatment to the requirements (regenerability) of a given used sand. It is generally assumed that the abovementioned portion of mass is subjected to treatment for a period of about 30 s. During the treatment process, the batch is put into rotation (400 rpm), forcing centrifugal movement of the regenerated sand onto the outer peripheral side surface of the batch made of an abrasion-resistant ceramic material. The dominant regenerative operations carried out in the device are abrasion, crushing of sand aggregates, and rubbing of the used coating from the surface of the grains. These operations are caused by obliquely arranged rollers, and the intensity of the regenerative effect can be changed by adjusting the pressure of the rollers against the batch.

As stated earlier, the side surface of the regenerator batch is made of an abrasion-resistant material. However, long-term tests of the device’s workings have shown that this material is in fact subject to abrasion. The results which demonstrate this are shown in Figure 1. The wear of the side surface material lowers the effectiveness of the reclamation process. This unfavorable effect can be partially decreased by reducing the distance between the pressing roller and the side of the device batch.

The confirmation of the impact of side surface material wear on the quality of the regenerate are the results of long-term tests of the device presented below. During the 2000 h measuring cycle, the device settings and the binding materials used for making molds did not change. The sand supplier was Sand Mine Szczakowa (Jaworzno, Poland), the producer of high quality sands for casting production.

In the long-term tests, together with the measurement of the maximum depth of side surface groove, shown in Figure 1, each time, the ignition loss of the used sand and the regenerate recovered from it were determined. The tests were carried out during the foundry’s production activity, where ALPHASET (Phenolic No-Bake Resin System) resin with the trade name Momentive TPA 70 (Momentive Performance Materials Italy S.r.l, Varese, Italy), hardened with ACE 1035 (Momentive Performance Materials Italy S.r.l, Varese, Italy), was used as the molding sand binding material. During the preparation of the molding sand, the resin addition was 1.1% in relation to the matrix, while the hardener addition was 0.35%. A mixture of 50% quartz sand with a grain size *d*_50_ = 0.32 mm and 50% regenerate obtained from the tested device was used as the matrix. The technology used in the foundry was designed in such a way that for most components, the castings are made in molds with an average ratio of molding sand to metal mass of 3.9:1. The foundry produces steel castings, usually the temperature of pouring molds with liquid metal is in the range of 1550–1600 °C. In the tests, as an indicator of the degree of regeneration of the molding sand, the ignition loss was used, which in the case of molding sand with an organic binder allows the determination of the residue of the binder on quartz sand grains. Ignition loss tests were carried out at a temperature of 950 °C, and five samples were tested in parallel, based on which the average value of the measurement was determined. At the start of the study, the regenerate’s ignition loss was 1.24%. The obtained results recording changes in ignition loss of the used sand and regenerate are presented in Figure 2. 

Based on the data presented in Figure 2, it can be seen that the achieved values of ignition loss of the used sand in the analyzed period of time were in the range of 2.43–2.55%; a similar analysis carried out for the regenerate indicates values of ignition loss in the range of 1.24–1.51%. The achieved values of material ignition loss after the regeneration process indicate a very high efficiency of the regeneration device operation. However, the average values of this parameter increase as a function of time. Assuming unchanged parameters of the process of making the mold and its overheating after the process of pouring with a liquid casting alloy, it can be assumed that the reason for the increase in the average value of the ignition loss is the decrease in the regeneration efficiency resulting from the change in the shape of the side surface (Figure 1).

This fact is confirmed by the data presented in Figure 3, which summarizes the average regeneration efficiency values expressed by Equation (1) for periods of regeneration device operation time: period 1 (0–500 h), period 2 (500–1000 h), period 3 (1000–1500 h), period 4 (1500–2000 h):(1)$$E=\left(1-\frac{LOI(R)}{LOI(US)}\right)\times 100\%$$
where:
*E*—regenerator’s work efficiency (%)*LOI(US)*—ignition loss of used sand (%)*LOI(R)*—ignition loss of regenerate (%).

Figure 4 presents an example photo of the morphology of the surface of used sand grains and the regenerate obtained after 2000 h of operation time of the regeneration device. The image shows a thick layer of used binding material covering the grain of the used sand and a much smaller amount covering the grain of quartz sand after the process of mechanical regeneration.

The conclusions resulting from the preliminary and industrial tests presented above indicate the need to develop a method enabling such control and setting of working elements of the USR-I device so that it is possible to eliminate the unfavorable effect of wear of these elements, causing a decrease in the quality of the regenerate.

## 4. Defining Problem and Purpose of Model Tests

The model tests were aimed at obtaining practical conclusions for determining the pressing force of the regenerator working elements on the processed grain of used sand, depending on the degree of wear of the regenerator batch.

In the steady state of device operation, the rollers rotate around the stationary axes, but relative to the side surface they roll over a layer formed of grains of the regenerated sand. This causes the displacement of sand under the rollers and the grains slipping on the side of the regenerator batch, which contributes to the formation of a gradually deepening groove of a circular cross-section (Figure 1). The wear of the side surface makes movement of the sand grains relative to each other difficult. For the regeneration process to take place with sufficient efficiency, it is necessary to gradually increase the pressing force of the rollers against the sand layer along with the increasing depth of the groove. If the force is too low, the grains will not be cleaned from the binding material envelopes, while if it is too high, sand grains may be ground.

The main goal was to build a calculation model that allows determining the optimal, from the point of view of the quality of the regeneration process, supply pressure in the pipe depending on the depth of the groove describing the wear of the side surface of the regenerator batch.

## 5. Regenerator Working Parameters—Preliminary Assumptions

Figure 5 is a diagram of the regeneration device from Omega Foundry Machinery Ltd. (Peterborough, UK), whereas Figure 6 shows a cross-section of the device with the marked dimensions of its components.

The basic geometrical data (Figure 6) and working parameters of the regeneration device are listed below:–roller diameter: *Z* = 240 mm–roller height: *H* = 65 mm–maximum radius of the side surface curvature: *R* = 283 mm–minimum radius of the side surface curvature: *r* = 267 mm–average radius of the side surface curvature: *R** = 275 mm–gap thickness between the roller and the side surface of the regenerator batch: *d* = 8 mm–rotational speed: *n* = 400 rev/min–regeneration device efficiency: *W* = 4700 kg/h = 78 kg/min = 0.195 kg/rev–regenerated sand density: *ρ* = 2300 kg/m^3^

It can be estimated that under steady state operation with the above parameters, each grain passes about 10–11 times under each roller. Based on the dimensions presented in Figure 6, the relationship between the force ***N*** pressing the roller to the side surface through a layer of the regenerated sand and the force ***F*** acting on the lever arm and resulting from the air pressure in the supply line can be determined. The leverage ratio is: *e* = *l*_2_/*l*_1_ = 639/162 = 3.94, therefore: ***N*** = *e*·***F*** = 3.94·***F***.

For the known diameter of the pneumatic hose (equal to 40 mm), the relationship between the regenerator supply pressure *p* and the pressing force ***N*** was determined: the pressure *p* in the hose equal to 1 bar corresponds to a pressing force ***N*** equal to 495 N (in further calculations, the rounded value 500 N was used). For the considered regenerative device, the minimum (initial) value of the supply pressure is equal to *p*_min_ = 2.4 bar, whereas the maximum value equals *p*_max_ = 4.2 bar. The minimum value of the roller pressure on the regenerated sand layer is ***N***_min_ = 1188 N.

It should be emphasized that the purpose of the work is not to accurately simulate the phenomena occurring during the regeneration process of the material obtained from used molding sands, but to simplify the analysis of the sand layer deformation process, which would allow the definition of a simple, phenomenologically justified indicator that can be used to assess the quality of regeneration.

The first assumption is the omission of inertia forces due to the fact that it can be estimated and justified that these forces have a negligible influence on the stresses in the sand layer in the domain close to the area of contact with the roller. When building the deformation calculation model, it was also assumed that despite a relatively high graininess, the layer of regenerated molding sand would be treated as a continuum.

From the modeling point of view, the problem is analogous (despite the small scale) to the problems of soil mechanics. It is necessary to adopt a constitutive model describing the elastic-plastic behavior of sand. The simplest model is described by the Mohr–Coulomb condition on which many other, more complex phenomenological models are based. In the discussed issue, the material model defined by the Drucker–Prager yield surface [24,25,26], which is often used for geological materials such as soils, rocks and sands, was adopted to describe the deformation process.

In loose materials, such as soil and sand, it is very difficult to experimentally determine physical constants (Young’s modulus, Poisson’s ratio), which, additionally, often prove to be dependent on pressure. In the simulations carried out, it was assumed that the physical parameters describing the sand are constant and do not depend on pressure.

## 6. Drucker–Prager Elastic–Plastic Model

The three-dimensional Drucker–Prager criterion allows the determination of the stress state at which material failure or plasticization occurs. This criterion is based on the assumption that at the moment of failure (plasticization), octahedral tangential stress is linearly dependent, through material constants, on the octahedral normal stress [6]. The D–P criterion can be presented as a yield function of the stress tensor $\tilde{\sigma }$ in the following pressure-dependent form:(2)$$f(\tilde{\sigma },\kappa )=\sqrt{J_{2}}+\alpha I_{1}-\kappa =0$$
where *I*_1_ is the first invariant of the stress tensor, *J*_2_ is the second invariant of the stress deviator tensor given by:
(3)$$I_{1}=\sigma_{1}+\sigma_{2}+\sigma_{3}$$
(4)$$J_{2}=\frac{1}{6}\left[{\left(\sigma_{1}-\sigma_{2}\right)}^{2}+{\left(\sigma_{2}-\sigma_{3}\right)}^{2}+{\left(\sigma_{3}-\sigma_{1}\right)}^{2}\right]$$

In the above formulas, *σ*_1_, *σ*_2_, *σ*_3_, denote principal stresses, and *α* and *κ* are material constants (*κ* is the yield stress under pure shear). The Drucker–Prager criterion can be considered as a special case of the Nadai criterion, or it can also be treated as an extension of the Huber–Mises criterion.

The D–P criterion describes a cone (for *α* > 0) or a cylinder (for *α* = 0) in the 3D space of principal stresses. Material constants are usually determined based on the results of standard three-axis compression and can be expressed in terms of the angle of internal friction *φ* and cohesion *c*.

The Drucker-Prager yield surface can be considered as a smooth version of the Mohr–Coulomb yield surface. It can be assumed that the Drucker–Prager yield surface circumscribes, inscribes or middle circumscribes the Mohr–Coulomb yield surface. After adopting the last of these cases, that the D–P failure cone passes through the interior vertices of the M–C hexagonal pyramid, the following expressions (taken in the numerical calculations) are obtained:(5)$$\alpha =\frac{2\cdot \sin\varphi }{\sqrt{3}(3+\sin\varphi )},\;\kappa =\frac{6c\cdot cos\varphi }{\sqrt{3}(3+\sin\varphi )}.$$

The main advantages of the D–P criterion are its simplicity and smooth, symmetrical yield surface in a stress space facilitating numerical implementation.

In the case of deformation of the regenerated sand layer presented in the paper, the plasticity D–P model described by the associated flow rule without hardening was adopted [26]. The constitutive equations of elastic-plastic deformation process are given in Appendix C.

## 7. MPM—Material Point Method

The finite element method (FEM), most commonly used in engineering analyses, is not effective enough for large deformation problems formulated in Lagrange’s description. Large mesh distortions progressing with body deformation are the cause of large errors of the approximated solution. Rebuilding the mesh is time consuming and introduces additional errors resulting from the need to project a solution from a distorted mesh onto a regenerated mesh. In the issue considered in this paper, an additional problem is the changing common contact surface of the roller and the penetrated sand layer (contact issue).

One of the methods used in cases of large deformation problems, for example in soil mechanics and flow problems of loose materials, is the material point method (MPM) [27,28]. This method, known in fluid mechanics as the particle-in-cell method (PIC), has been adapted to the problems of solid mechanics. One of the generalizations of the MPM method is the generalized interpolation material point (GIMP) method [29].

In the MPM and GIMP methods, two types of discretization are used. The first discretization type (related to the description in Lagrange coordinates) consists of dividing the region occupied at the initial moment by the body in question into a set of sub-regions, each represented by one particle (material point). The second type of space discretization (referring to the description in Euler coordinates) is associated with the so-called computational mesh covering possible positions of the analyzed body.

MPM and GIMP owe their effectiveness to the solving equations describing a given problem using a computational mesh that can be modified arbitrarily during analysis or remain unchanged, while state variables are assigned to the moving Lagrange particles.

The standard MPM method treats each particle as a concentrated mass, and the GIMP method replaces calculation of integrals in MPM using the δ-Dirac function with weighted average integrals over a finite domain in the vicinity of the particle using a weight function, also called a characteristic function.

The basic difference between MPM and FEM is that in the former, state variables (i.e., mass, stress, strain, velocity, etc.) are assigned to material points that are defined independently of the computational mesh. In the latter, state variables are specified at integration points connected with the elements. More mathematical details of the MPM are presented in Appendix D.

## 8. Material Constants—Experimental Research Results

The coefficients describing the constitutive model of sand adopted in numerical calculations were determined based on the results of experimental research conducted on sand used in the foundry. Standard triaxial compression, direct shear and odometer tests were carried out, and the elastic-plastic model parameters obtained are given in Table 1.

It should be emphasized that for loose materials, the results of experimental tests describing the elastic behavior are always associated with large errors, hence the significant variability of the Young’s modulus.

The following values were adopted for numerical simulations: *φ* = 35°, *c* = 3 kPa. Despite the fact that for sand, the Young’s modulus depends on pressure, a constant value was used in the simulations: *E_S_* = 87.0 MPa.

Details on experimental testing of the properties of regenerated molding sand are presented in Appendix E, where Figure A7, Figure A8, Figure A9 and Figure A10 show a sample after test carried out in the triaxial compression apparatus and charts with the exemplary results obtained.

## 9. Geometric Model of System

The built-up geometric model consists of three solids representing a roller, a layer of sand and a part with the side surface of the batch of the regeneration device (Figure 7).

As the pressure exerted by the roller is located on a relatively small area, there is no need to consider a layer of sand spread over the entire side of the regenerator batch. It was assumed that the layer of sand is spread over a side segment with an angular width of 40 degrees. Instead of the entire roller, its cross-section was formed the upper part of which is a rectangle measuring 212 mm × 65 mm.

The built-up CAD model is parametric, allowing, among others, continuous change in the sand layer thickness and the sidewall groove depth, thus enabling a description of its progressive wear. The thickness of the layer describing the side surface before wear is 20 mm, while its width and the width of the regenerated sand layer are assumed to be equal to 110 mm.

Figure 7 shows the model without side surface wear. In Figure 8, the side surface has a groove with a maximum depth of 5 mm. The cross-section of the groove has a circular shape with rounded edges. The groove shape was modeled on the basis of the observation of the worn surface of the reclamation device (Figure 1c).

## 10. Assumptions of Computational Model Description in MPMsim

In the case of deformation of the regenerated sand layer presented in the paper, the Drucker–Prager plasticity model together with the associated flow law without hardening was adopted (Appendix C). In the numerical calculations, it was assumed that the deformation of the sand layer under the pressure exerted by the roller is a static process, and the inertial forces resulting from the movement of the roller in the sand layer were omitted. In addition, the lack of friction between the sand and the roller and between the sand and the side surface of the regeneration device was assumed.

Numerical simulations were made using the MPMsim program which employs the material point method (www.mpmsim.com). The built CAD model was imported into the MPMsim program in which displacement constraints were imposed on the lower part of the side surface fragment. An evenly distributed normal load was applied to the pressure roller cross-section area. This load generates a force pressing the roller against the sand layer, depending on the air pressure in the supply line. Figure 9 shows the diagram in the MPMsim program of the considered system with the applied load.

In the computational model, it was assumed that the roller and the side surface are made of an elastic material with a Young’s modulus $E=3.6\times 10^{11\;}\mathrm{Pa}$ (and with a Poisson’s ratio of 0.27), i.e., over 4000 times larger than the Young’s modulus determined experimentally for the tested sand. With such a high ratio of Young’s modulus, the roller and side surface of the regenerator batch deformations are negligible compared to the deformations of the sand layer.

For the Drucker–Prager plasticity model, additional parameters describing the behavior of the material can be given: the angle of dilation and tensile strength.

The angle of dilation ψ is responsible for the change in volume occurring during plastic shear. It is usually assumed that its size is constant. The condition ψ=0 means that the volume does not change during plastic shear deformation. Clays have very low expansion (ψ≈0), while for sands the angle of dilation depends on the angle of internal friction [30].

For sands and gravels with the internal friction angle φ>30°, the angle of dilation can be estimated using the expression ψ=φ−30°. A negative value of the angle of dilation is only allowed for very loose sands. In most cases, the condition assumed is ψ=0.

The physical phenomenon of coherence responsible for the non-zero value of cohesion *c* is also responsible for the finite (usually negligible) strength of the geological material (soil, sand) to tensile stress. The tensile strength is usually a fraction of the cohesion value *c*.

Numerical calculations assume that the angle of dilation is 5 degrees, while the tensile strength is assumed to be equal to 1 kPa. In the calculation algorithm used by the MPMsim program, this parameter is associated with the possibility of reaching a limit state by exceeding the allowable tension value (the so-called ‘tension cut-off’).

## 11. Quality Indicator of Sand Regeneration

The purpose of the numerical simulations was to investigate the course of the deformation process of the sand layer located between the roller and the side surface of the regeneration batch and, on this basis, to define the simplest possible indicator allowing comparison of the degrees of regeneration of the used molding sand achieved for different values of air pressure in the supply line.

In order for the regeneration process to take place, sufficiently large shear deformations in the regenerated sand layer, which occur at a sufficiently high stress level, are necessary. Even large stresses occurring in an area in which, for example, boundary conditions prevent deformation will not cause particles of sand to move relative to each other and deformation will be small. On the other hand, large shear strains at low stresses will also not be suitable conditions for sand regeneration.

It is difficult to theoretically determine the optimal conditions under which the regeneration process should occur and, what is more, it would require a number of experimental studies. On the basis of the results of the regeneration process taking place in the tested device, however, it is possible to describe the conditions in which regeneration occurs with optimal efficiency.

It is known that for the device considered in the paper that does not yet have wear of the side surface (depth of the groove equal to *h* = 0 mm), the optimal regeneration conditions are ensured by the supply air pressure *p* = 2.4 bar (pressing force ***N*** = 1200 N). It is also known that for a maximum depth of the groove equal to *h* = 5 mm, optimal regeneration conditions occur for the supply pressure equal to *p* = 3.6 bar (pressing force ***N*** = 1800 N). This pressure value was determined experimentally in such a way that the values of ignition losses were comparable in these two cases which are reference systems, on the basis of which a numerical indicator of regeneration quality will be defined, whose value for these two systems should be with high accuracy almost the same.

The cross-sections visible in Figure 10 and Figure 11 show the deformation of the sand layer (middle section) for two reference states intended to define the regeneration quality indicator. The figures show the displacement in the *y* direction, parallel to the axis of the roller.

Based on the observation of the results of the numerical simulations of sand layer deformation, as the indicator of the quality of regeneration, marked by *Q,* was taken the maximum displacement of the middle section of the layer in the *y* direction, parallel to the axis of the roller, marked as $y_{\max}$, divided by the maximum thickness of the sand layer before its deformation, marked by $\delta_{\max}$:(6)$$Q=\frac{y_{\max}}{\delta_{\max}}$$

The adopted quality indicator *Q*, which is in a sense an approximation of the average shear strain component, may be considered as a function of the pressure *p* (bar) and the maximum depth of the groove *h* (mm): Q=Q(p,h). For the unused side surface (*h* = 0 mm) and pressure *p* = 2.4 bar, the quality indicator is: $Q_{1}=Q(2.4,0.0)=0.024985$, while for the worn side surface with a groove depth equal to *h* = 5 mm and pressure *p* = 3.6 bar is obtained: $Q_{2}=Q(3.6,5.0)=0.024402$.

The values $Q_{1}$ and $Q_{2}$ differ by approximately 2%. As the optimal indicator was arbitrarily adopted the approximate arithmetic mean of the values $Q_{1}$ and $Q_{2}$: $Q_{OPT}=0.024700$.

It was assumed in the calculations that the thickness of the sand layer for the unused side surface before the displacement of the roller is 9 mm, i.e., in this case $\delta_{\max}=9.0+h=9.0+0.0=9.0\;\mathrm{mm}$ is obtained, while for the worn side surface with a groove depth of *h* = 5 mm, $\delta_{\max}=9.0+h=9.0+5.0=14.0\;\mathrm{mm}$ is obtained. For the value of the original layer thickness adopted in this way, the minimum displacement of the outer surface of the roller (i.e., its part nearest to the side surface) in a direction perpendicular to its axis at supply pressure *p* = 2.4 bar is approximately 1.5 mm (exact simulation result: 0.00164 m), i.e., after depressing the roller, the minimum thickness of the sand layer is approximately 7.5 mm (exact value from the simulation: 0.00736 m). This is the approximate width of the gap between the roller and the side surface at steady state of the regeneration device for the unused batch.

## 12. Results of Numerical Calculations

Based on the defined quality indicator *Q*, it is possible to estimate the optimal supply pressure value in the system for side surface groove in the range: 0.0 mm<h<5.0 mm. After exceeding the upper value of the range, the ceramic coating of the side surface of the regenerator batch is practically suitable for replacement.

For the groove depth described by the values h=1.0, 2.0, 3.0, 4.0 mm (respectively obtained at $\delta_{\max}=10.0,\;11.0,\;12.0,\;13.0\;\mathrm{mm}$) such a value of the force ***N*** (and also the supply pressure *p*) is searched for that presses the roller against the sand layer, for which the deformation will be described by the value of the quality indicator *Q* not very different (with an error of not more than 1%) from the value $Q_{OPT}=0.024700$.

The graph in Figure 12 (and data collected in Table 2) shows the value of the optimal air supply pressure depending on the depth of the side surface groove. It can be seen that the optimum pressure value depends approximately linearly on the value of the groove depth.

Figure A11, Figure A12, Figure A13 and Figure A14 (Appendix F) show the displacements in the sand layer in *y* direction, parallel to the axis of the roller, for the different values of the side surface groove and for the optimal air pressure values in the supply line.

Figure 13, Figure 14 and Figure 15 present graphs of, respectively: the maximum displacement in the middle section layer in the *y* direction parallel to the axis of the roller; displacement of the roller (its part nearest to the side surface) in a direction perpendicular to its axis; indicator of the quality of regeneration depending on the pressure in the supply line for the maximum groove of the side surface (described by the value *h* = 5 mm).

The graphs presented in Figure 13, Figure 14 and Figure 15 reveal that for the quantities given, their variability is a non-linear function of the supply pressure, although in the tested pressure range the deviation from linearity is small. Increasing the pressure would surely give graphs that differ more and more from linear functions, although above a certain pressure value the grains of sand would start to be destroyed, which would change the parameters describing it, such as the Young’s modulus and Poisson’s ratio. This would require additional experimental testing which would not be justified from a practical point of view, because the goal of the regeneration process of molding sand is to remove the resin without destroying its quartz grains.

## 13. Verification Tests Results

In order to confirm model calculations, which showed that with a certain wear level of the peripheral side surface of the regeneration device, it is possible to increase the regenerative impact by increasing the supply pressure value of the cylinders pressing the roller to the side surface, verification tests were carried out.

Verification tests were carried out using the USR-I device, which was excluded from production for their duration. They consisted of subjecting a portion of the used sand, after the initial regeneration process, consisting of crushing lumps and clumps of grains, to the regenerative effect in the USR-I device operating at various settings of pressure supplying the cylinders pressing the rollers to the side surface.

The tests were carried out after a device operation time of 2000 h, at which the groove depth of the side surface was 3.8 mm (Figure 1). The following supply pressure values were used: 2.5 bar, 3.0 bar, 3.5 bar, 4.0 bar. Each portion of sand subjected to regeneration was 30 kg, while the regeneration time was 30 s. Due to the fact that a periodic mode of the device operation was used, it was possible to classify and dedust regenerates obtained for individual device settings.

During the tests, the following parameters of the used sand and regenerate were determined: ignition loss, pH, grain size *d*_5__0_, and the amount of dust generated in the regeneration process. The results of the tests are presented in the set of charts in Figure 16.

The results of the ignition loss of regenerates obtained after various impact of the regenerator, measured by the values of supply pressure, which are presented in Figure 16a, show that as the value of supply pressure increases, a decrease in the value of the ignition loss of obtained regenerates is observed. A similar effect is observed by analyzing the results of the pH of obtained regenerates, which, together with the increase in the intensity of the regenerator’s impact, decrease, aiming at the reaction obtained for fresh sand (Figure 16b).

The grain size *d*_50_, for which the results are presented in Figure 16c, also decreases as a result of the increase in supply pressure. At this point, it should be noted that quartz sand, characterized by its diameter *d*_50_ = 0.32 mm, is used to prepare the molding sand. Therefore, obtained in the verifying tests, the results of the analysis of regenerate grain size which show a smaller diameter than that specified for fresh sand may indicate too intensive impact of the regenerator and a progressive, unfavorable phenomenon of crushing of quartz matrix grains. This effect was obtained for the regeneration process carried out at a supply pressure of 4 bar.

Figure 16d presents the results of the amount of dust generated in the regeneration process. It was possible to conduct this test owing to the use of a periodic device operation mode. Thanks to that, after the regeneration of 30 kg of used sand, it was possible to carry out a pneumatic classification and determine the amount of dust generated in the process. These dusts were created as a result of elementary rubbing, abrasion and crushing operations taking place in the device (Figure A3, Figure A4 and Figure A5). A steady increase in the amount of dust can be seen up to 3.5 bar, after regeneration at a supply pressure of 4 bar, a step increase in the amount of dust has been observed.

Comparing this with the value of the dust ignition loss from Figure 16e, showing the amount of organic substance it contains (binders removed from the surface of the matrix grains as a result of the regeneration process), it can be stated that the regeneration process carried out at the value of the supply pressure of the roller cylinders exceeding 3.5 bar leads to the adverse effect of crushing of quartz matrix grains.

## 14. Discussion

The results of the experimental tests presented in Section 13 are consistent with the results of numerical calculations presented in Section 12. The diagram in Figure 12 shows that the optimal pressure for groove depth *h* = 5 mm is equal to *p* = 3.3 bar. By approximating the given values from Figure 16a, the ignition loss for pressure *p* = 3.3 bar equals 1.29, i.e., the value is slightly different from the value obtained for *h* = 0 mm and *p* = 2.4 bar, equal to 1.24. Therefore, the introduced quality indicator *Q*, based on the quantities related to shear deformation of the sand layer, can be associated with the ignition loss of the regenerated sand.

The regeneration quality indicator defined in the work, associated with the shear deformation in the middle section of the sand layer, justified by the physical course of the regeneration process, allowed the determination of the optimal air supply pressure of the centrifugal device, depending on the degree of wear of the side surface of the regenerator batch, described by the depth of the groove formed in it.

It is worth emphasizing that the presented results of the calculations of the optimal air pressure relate to a specific regenerative device and are valid for specific parameters, such as the diameter of the pneumatic hose, leverage ratio and given dimensions of the roller and batch.

Based on the conducted verification tests, correctness of the developed model and compliance of the test results obtained with the parameters of devices regulated on the basis of the results of numerical calculations were confirmed.

The developed theoretical model allows the optimization of the industrial device operation while maintaining the required reclaim quality determined by the range of loss on ignition as well as the size and shape of quartz basis grains. The model calculations will enable the development of the device’s automated work system responding to the emerging effect of wear on the side surface by means of changing the distance (via controlling the supply air pressure) between and mutual arrangement of the cooperating mechanical system components, the roller and the side surface of the batch.

## 15. Conclusions

The simulation and experimental studies were carried out for a specific regenerative machine and a specific molding sand described in the publication. However, the qualitative conclusions that can be drawn from the results obtained can be used for another regenerative machine of similar type with different dimensions and for sand with different physical parameters. Resting upon the implemented regeneration quality index based on the deformation of the sand layer occurring in the regenerator, the optimal machine process parameters, determined by the supply pressure (i.e., the optimal regenerator roller pressing force on the layer) and depending on the degree of side surface wear (given by the maximum groove depth), can be obtained. An approximate linear relationship was obtained for the case examined in the paper, but the result does not always have to be identical. Therefore, the supply pressure can be changed depending on side surface wear to ensure adequate quality of the obtained regenerate, and the method of pressure control can be obtained still prior to starting the regeneration process by virtue of anticipating a change in geometry of the side surface as it wears.

## Figures and Tables

**Figure 1 materials-13-02134-f001:**
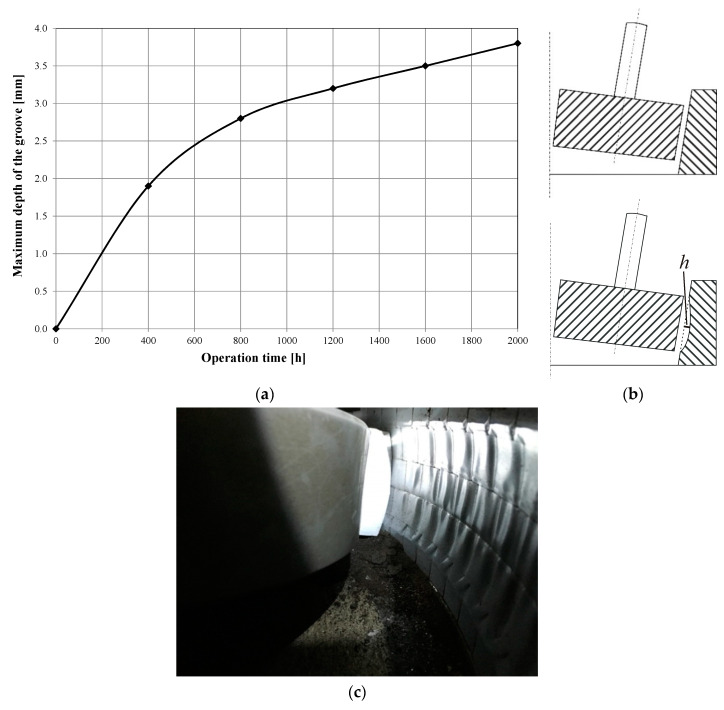
Wear of regenerator batch lining as a result of long-term operation, measurement at constant supply pressure set as *p* = 2.4 bar: (**a**) maximum depth of side surface groove (marked by *h* in drawing (**b**)) as a function of regenerator work time; (**b**) scheme of regenerator batch lining wear; (**c**) photo showing regenerator batch lining wear.

**Figure 2 materials-13-02134-f002:**
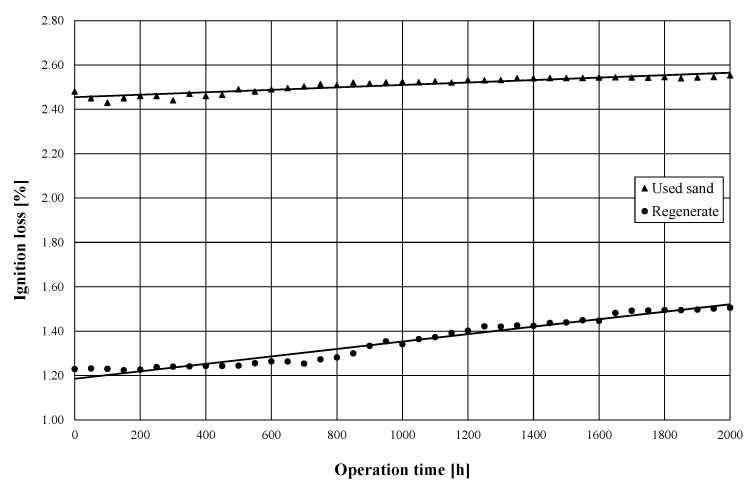
Ignition loss of used sand and regenerate as a function of USR-I device operation time, measurement at constant supply pressure set as *p* = 2.4 bar.

**Figure 3 materials-13-02134-f003:**
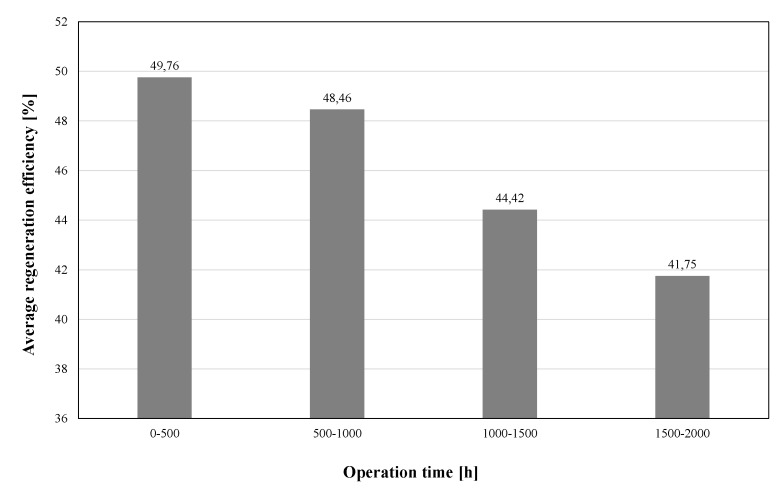
Average efficiency of the USR-I regenerator demonstrated in long-term studies.

**Figure 4 materials-13-02134-f004:**
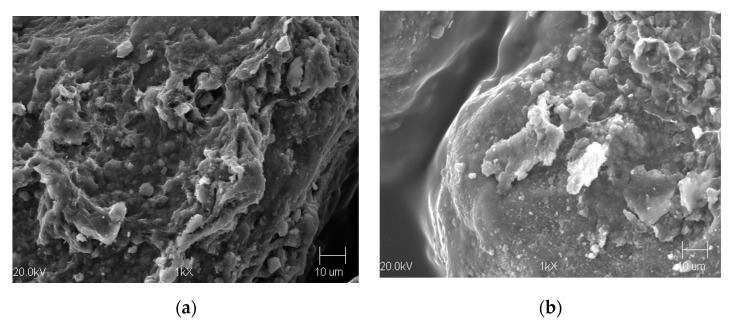
Surface morphology of used sand grains (**a**) and regenerate (**b**) obtained after 2000 h of operation time of the USR-I device; 1000× magnification.

**Figure 5 materials-13-02134-f005:**
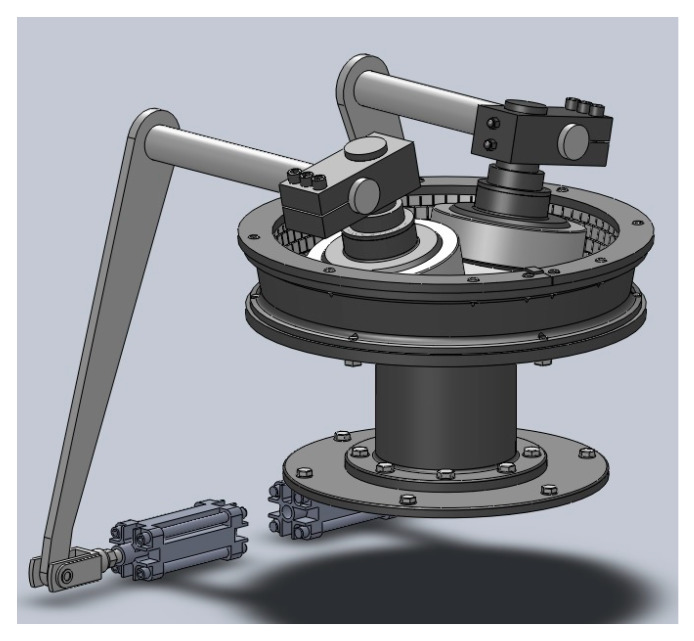
Diagram of regeneration device from Omega Foundry Machinery Ltd.

**Figure 6 materials-13-02134-f006:**
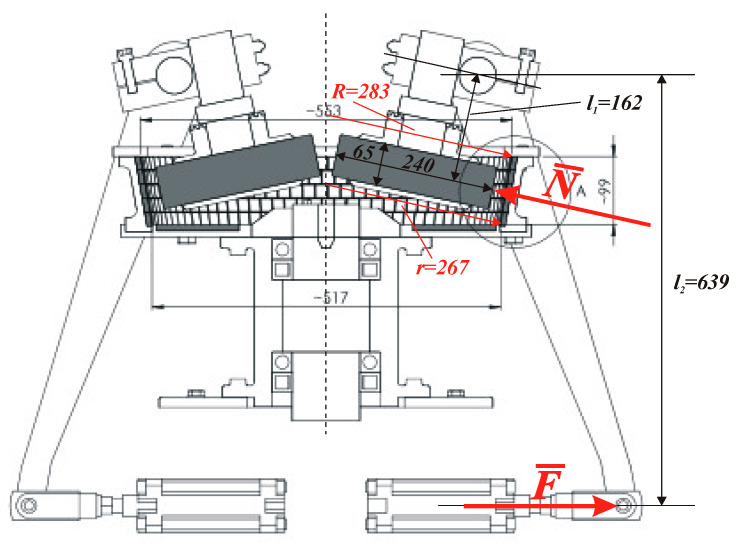
Cross-section of regeneration device.

**Figure 7 materials-13-02134-f007:**
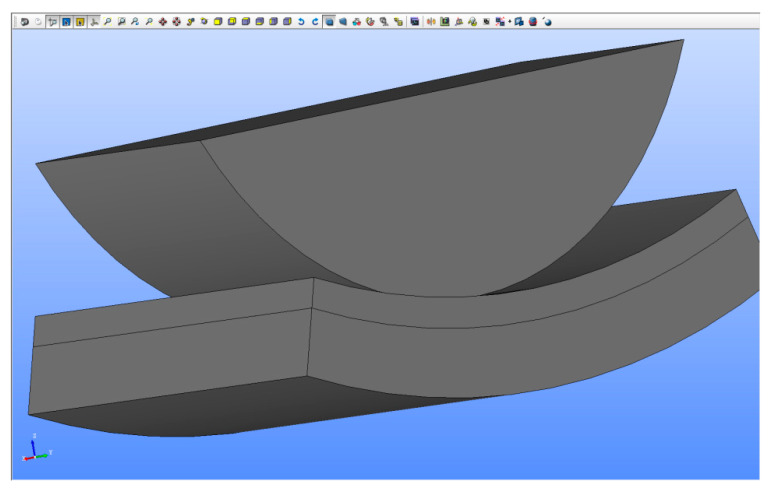
System CAD model—side surface of regenerator batch not worn.

**Figure 8 materials-13-02134-f008:**
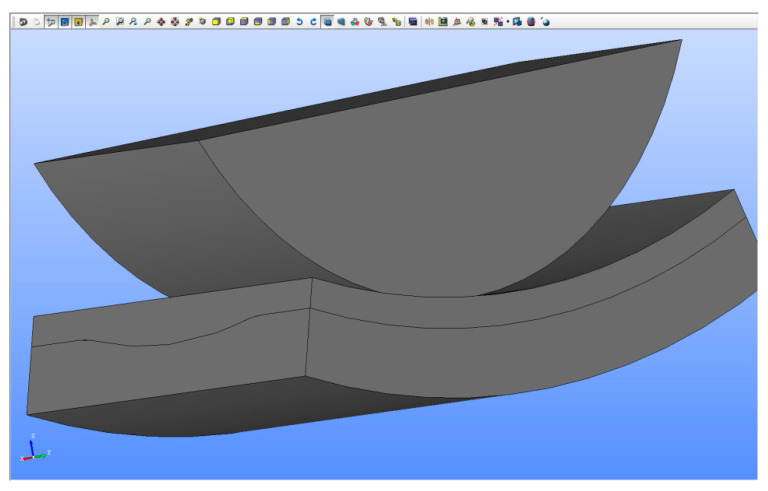
System CAD model—side surface of regenerator batch worn to depth of 5 mm.

**Figure 9 materials-13-02134-f009:**
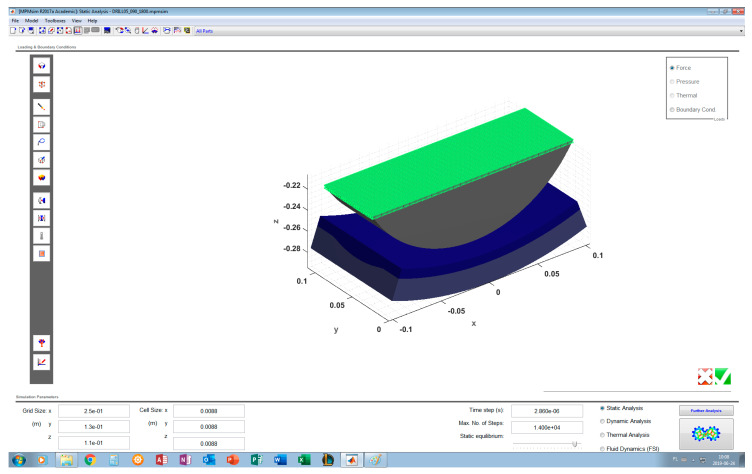
Computational model in MPMsim—normal load of roller cross-section.

**Figure 10 materials-13-02134-f010:**
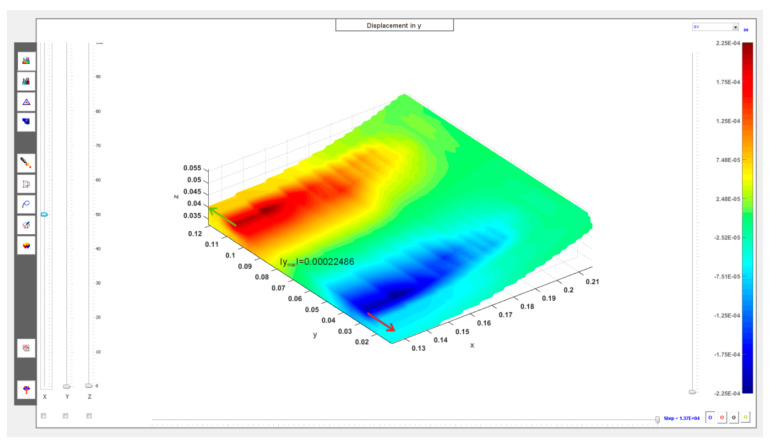
Displacements (in meters, middle section) in sand layer in parallel *y* direction to roller axis: *h* = 0 mm, *p_opt_* = 2.4 bar (***N*** = 1200 N).

**Figure 11 materials-13-02134-f011:**
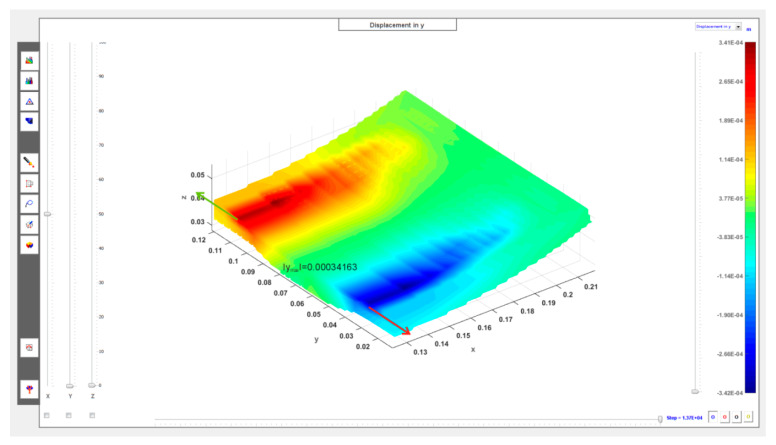
Displacements (in meters, middle section) in sand layer in parallel *y* direction to roller axis: *h* = 5 mm, *p_opt_*= 3.6 bar (***N*** = 1800 N).

**Figure 12 materials-13-02134-f012:**
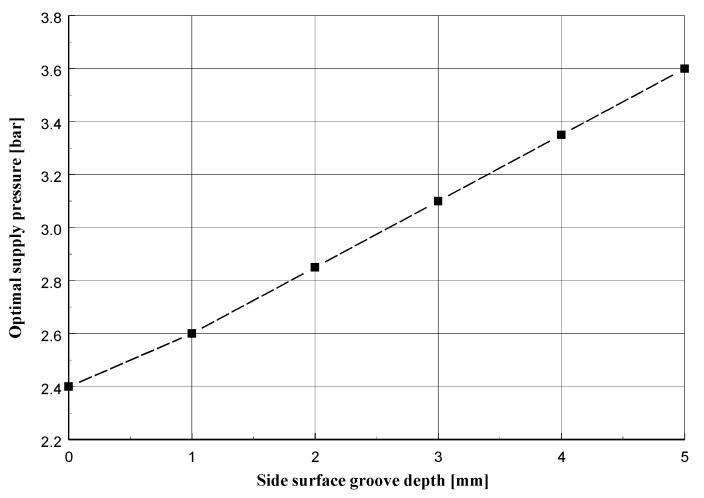
Optimal supply pressure *p_opt_* depending on depth of side surface groove *h*.

**Figure 13 materials-13-02134-f013:**
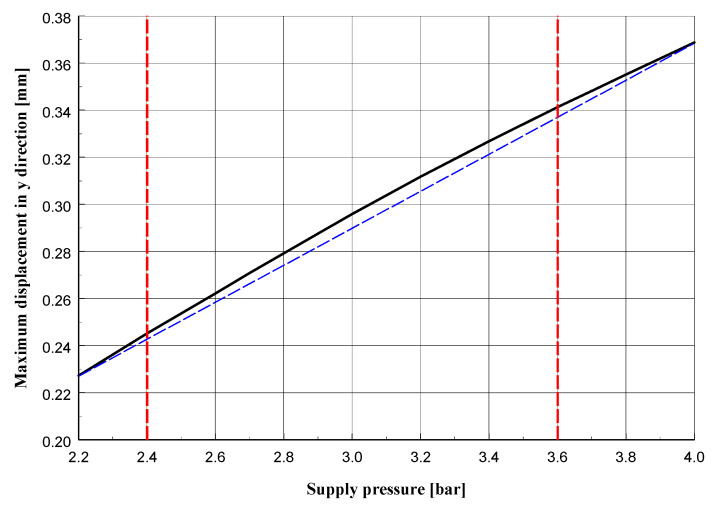
Maximum displacement in sand layer (middle section) in *y* direction parallel to roller axis depending on pressure in supply line.

**Figure 14 materials-13-02134-f014:**
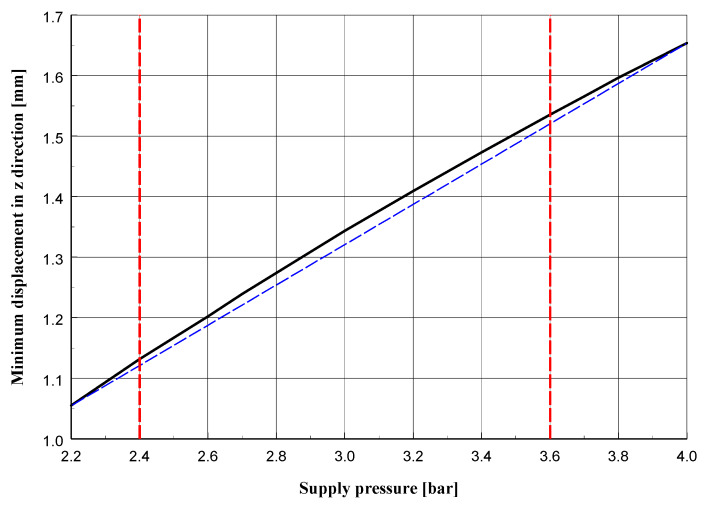
Displacement of roller (its part nearest to side surface) in direction perpendicular to its axis depending on pressure in supply line for *h* = 5 mm.

**Figure 15 materials-13-02134-f015:**
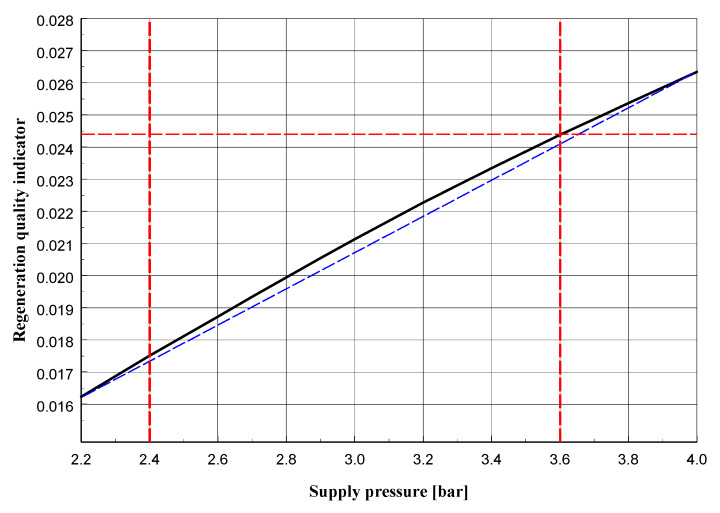
Regeneration quality indicator depending on pressure in supply line for *h* = 5 mm.

**Figure 16 materials-13-02134-f016:**
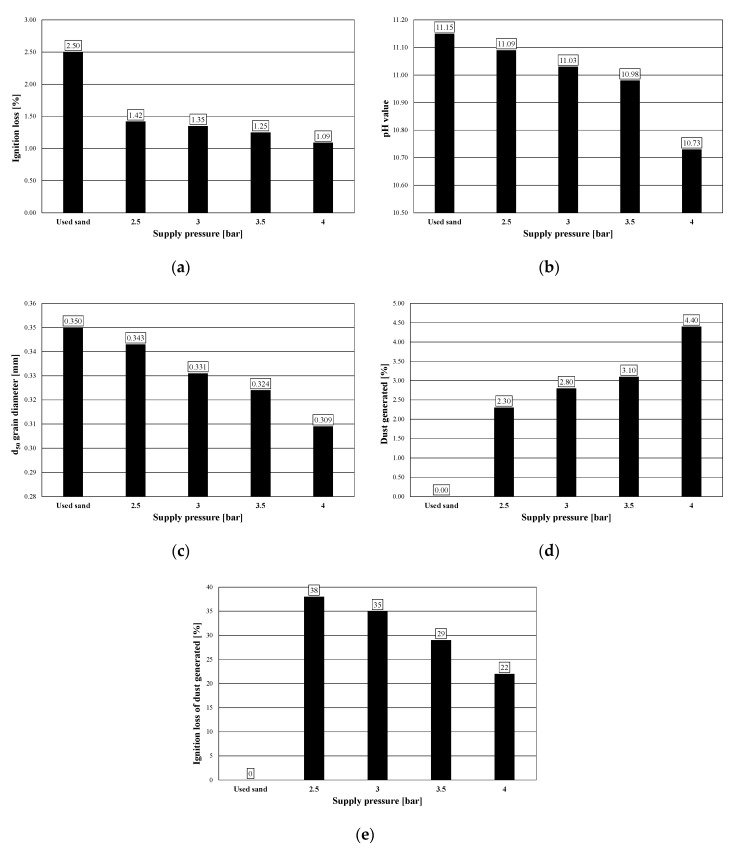
Results of verification tests: (**a**) ignition loss of obtained regenerates; (**b**) pH of the obtained regenerates; (**c**) regenerate grain size *d*_50_; (**d**) amount of dust generated in the regeneration process; (**e**) ignition loss of dust.

**Table 1 materials-13-02134-t001:** Physical parameters (elastic-plastic model) of regenerated sand.

Angle of Internal Friction	Cohesion	Poisson’s Ratio	Young’s Modulus
φ=32°−37°	c=0 kPa−4 kPa	ν=0.25	E=86.9±16.1 MPa

**Table 2 materials-13-02134-t002:** Optimal supply pressure (elastic-plastic model) of regenerated sand.

Maximum Depth of the Side Surface Groove *h* (mm)	Optimal Supply Pressure *p_opt_* (bar)
0.0	2.40
1.0	2.60
2.0	2.85
3.0	3.10
4.0	3.35
5.0	3.60

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
