# Peer review of "Application of 3-D Drucker–Prager Material Model to Determine Optimal Operating Parameters of Centrifugal Regeneration Device"

_materials, 2020, doi:10.3390/ma13092134_

Round 1

Reviewer 1 Report

In this paper the authors show a proposal for the improvement of an equipment for the recovery of foundry sands. The work makes a tour on the procedures and principles to recover casting sands. It then develops an analytical, elastic-plastic model and describes and applies the material point method (MPM) to simulate the wear of the rollers and the walls of the equipment.

From my point of view this work is badly structured and confusing. The authors should undertake modifications in:

  1. There is no section on material and methods, there is no section on conclusions. There are no detailed essays. Nothing is said about the properties of the sands used and their characteristics including types of binders, additives, moisture content, fineness index, etc. Nothing is said about the characteristics of the recovered sands. It is not known who the suppliers of the sands with their additives are. The time or volume of recovery of the equipment is not known. Have trials been carried out with the equipment, and could the authors describe to us what these trials have been like and what has been measured? Authors should remember that one of the objectives of any research is the possibility of replication.
  2. The authors set their objective on the improvements of a particular team and it is not known if there are other teams that solve the problem and, from my point of view, the work has not focused an analysis of the teams for this purpose.
  3. In section 9 the authors show us some constants obtained from an experimental trial. The trial, the equipment used, the type and number of samples, the error of assignment of the values, etc., are unknown.
  4. The discussion should provide numerical results and limit the results to the characteristics of the sands studied and the equipment used. The interest is very poor as finally only the study of the adequate pressure of the regeneration device has been obtained. In addition, it seems that the conclusions are based on the MPM model and it is not known if tests have been carried out to verify the goodness of the model.
  5. On the other hand, the work does not flow, is excessively extensive and is difficult to follow. The images used in some cases are of low quality as they are not well appreciated, figures 16, 17, 18, 19 and , on the other hand, figures 20, 21 and 22 seem to be excessive in size. It is at least disturbing that the authors did not have to use any tables in the manuscript.

Reviewer 2 Report

The introduction part of the work presents only very general aspects of the use of sand molds in the process of manufacturing cast metals. There are no references to papers tackling the same problem as the one targeted by the authors, regenerating the sand form sand molds. Moreover, in paragraph 2, references 2 and 3 are used repetitively, while they are belonging to one of the authors of the present paper. It is normal to show that the authors have experience in the field, but the first sections of the paper should present what others did in the field and what is to be done further, not to present general aspects and highlight the previous work of the authors.

What’s the purpose of presenting the USR-I centrifugal device so thoroughly in paragraph 3? Is it designed/build by the authors only for the experimental work presented in the paper? If yes, please highlight that. If it is a commercially available device, there is no use to present it in such details.

Lines 356-358: “Standard triaxial compression tests were carried out, fresh sand and regenerated sand were tested, and the following parameters were obtained” – some description of how were the tests carried out should be introduced in the paper, or, at least a reference to some works which describe how these tests are performed.

Line 361: “In addition, the following were obtained for both types of sand:” – how were those values obtained?

It is very unclear and hard to follow what is the significance of each curve from figures 20-22 (even it is briefly explained in text). Some notations on figures will be helpful.

Lines 538-540: “Based on the conducted verification tests, correctness of the developed model and compliance of the test results obtained with the parameters of devices regulated on the basis of the results of numerical calculations were confirmed.” – please elaborate how was that confirmed.

Round 2

Reviewer 1 Report

The authors have responded to most of the considerations indicated in the review. The work has been substantially modified. I thank them for their work.

Reviewer 2 Report

The authors have addressed al the issues pointed by the reviewer, in a logical and comprehensive manner.

It is also commendable that some non-critical information (yet needed in order to fully understand the approach) was moved to annexes.

In my opinion, the quality of the paper was improved significantly.

I consider that in this form the paper is worthwhile to be published in the journal.